# Imbalance of ER and Mitochondria Interactions: Prelude to Cardiac Ageing and Disease?

**DOI:** 10.3390/cells8121617

**Published:** 2019-12-12

**Authors:** Jin Li, Deli Zhang, Bianca J. J. M. Brundel, Marit Wiersma

**Affiliations:** Department of Physiology, Amsterdam UMC, Vrije Universiteit Amsterdam, Amsterdam Cardiovascular Sciences, 1081 HV Amsterdam, The Netherlands; d.zhang@amsterdamumc.nl (D.Z.); b.brundel@amsterdamumc.nl (B.J.J.M.B.)

**Keywords:** endoplasmic reticulum stress, mitochondrial stress, unfolded protein response, protein quality control system, cardiac ageing, cardiac disease, atrial fibrillation, cardiomyopathy

## Abstract

Cardiac disease is still the leading cause of morbidity and mortality worldwide, despite some exciting and innovative improvements in clinical management. In particular, atrial fibrillation (AF) and heart failure show a steep increase in incidence and healthcare costs due to the ageing population. Although research revealed novel insights in pathways driving cardiac disease, the exact underlying mechanisms have not been uncovered so far. Emerging evidence indicates that derailed proteostasis (i.e., the homeostasis of protein expression, function and clearance) is a central component driving cardiac disease. Within proteostasis derailment, key roles for endoplasmic reticulum (ER) and mitochondrial stress have been uncovered. Here, we describe the concept of ER and mitochondrial stress and the role of interactions between the ER and mitochondria, discuss how imbalance in the interactions fuels cardiac ageing and cardiac disease (including AF), and finally assess the potential of drugs directed at conserving the interaction as an innovative therapeutic target to improve cardiac function.

## 1. Introduction

Cardiac disease is still the leading cause of morbidity and mortality worldwide, even though the clinical management of cardiac diseases has been improved over the last years [1]. In particular, due to the ageing population, atrial fibrillation (AF), the most common cardiac tachyarrhythmia, shows a steep increase in incidence and, therefore, healthcare costs. Its prevalence in the developed world is estimated to be 1.5%–2% of the general population and steadily rises in the population between 75 and 85 years of age [2]. In Europe, more than 6 million people suffer from AF and in the next 50 years its incidence is estimated to at least double due to the ageing population [3]. In addition, heart failure (HF), the most common and chronic stage of cardiac disease, is also approaching a higher prevalence all over the world. HF constitutes a medical and economical problem worldwide, as treatment costs reach as much as 1%–2% of the total healthcare budget [4]. Unfortunately, no specific therapy exists for AF and HF, and, therefore, treatment is severely hampered. Reason for treatment failure is that the exact molecular mechanisms underlying the pathophysiology are still unknown. However, accumulating evidence shows that derailment of proteostasis (i.e., the homeostasis of protein expression, function, and clearance) is a central component driving cardiac ageing and cardiac disease [5,6,7].

A healthy proteostasis network safeguards normal cellular (metabolic) function through proper protein biosynthesis, folding, trafficking, and clearance, thereby maintaining normal heart function [5,8]. Proteostasis functioning is monitored by the protein quality control (PQC) system, which consists of the heat shock response (chaperones), unfolded protein responses, and protein degradation systems, including the ubiquitin–proteasomal system (UPS) and autophagy [9]. In general, cellular stress, including stress caused by cardiac disease, activates multiple stress pathways, including the cytosolic heat shock response, the endoplasmic reticulum (ER) unfolded protein response (UPR^ER^), and the mitochondrial UPR (UPR^mt^) [10,11,12].

Within the PQC, the ER and mitochondria play a critical role in the regulation of protein homeostasis and the maintenance of normal cellular function. The ER is an essential organelle involved in protein synthesis, folding, and trafficking. As protein folding is an error-prone process, the PQC system of the ER is specialized in optimizing this process, thereby preserving cardiac protein quality and homeostasis [13]. As approximately 30% of the cardiomyocyte volume is comprised of mitochondria, the maintenance of a healthy and functional mitochondrial PQC system, which includes molecular chaperones (e.g., HSP60 and HSP10) and proteases (e.g., ClpP), is critical to conserve the energy balance and cardiomyocyte function. The PQC system in mitochondria activates, upon stress, protein folding assistance mechanisms and promotes clearance of misfolded proteins to preserve mitochondrial function [14,15].

Thus, a healthy proteostasis in cardiomyocytes safeguards the proper contractile function in the heart. The ER and mitochondria are essential organelles for regulating protein homeostasis, thereby guaranteeing cardiomyocyte function and survival. Therefore, a deeper understanding of ER and mitochondrial function during health and cardiac disease is required to ultimately develop novel therapeutic strategies.

## 2. Role of the ER in Cardiac Health

The ER is an essential organelle supporting multiple cellular processes such as protein synthesis, protein folding, regulation of Ca^2+^ homeostasis, and contribution to the generation of autophagosomes and peroxisomes [16]. The ER lumen constitutes of a specialized folding environment, including ER chaperones and oxidoreductases, which processes approximately one-third of newly synthesized proteins, including secretory, cell membrane, and ER proteins, to ensure that they are correctly folded and assembled [17]. One of these ER chaperones is glucose-regulated protein 78 (GRP78), a member of the heat shock protein 70 family, which promotes the folding of hydrophobic stretches in newly synthesized polypeptides [18,19]. The oxidizing folding environment in the ER is vital for the disulfide bond formation catalyzed by protein disulfide isomerase (PDI) and ER oxidoreductin 1 (Ero1) [20,21].

Besides protein-related functions, the ER is also a high-capacity reservoir for intracellular Ca^2+^, which is important for the regulation of muscle contraction and relaxation [22,23]. For example, in cardiomyocytes, the sarcoplasmic reticulum (SR), a muscle-specialized form of the ER, regulates Ca^2+^ fluxes and controls the excitation–contraction coupling of the heart [24]. In the ER/SR, several major proteins are involved in Ca^2+^ uptake and release: Ca^2+^-ATPases (SERCAs), that transport cytosolic Ca^2+^ into the ER lumen, and the ryanodine (RyRs) and inositol 1,4,5-triphosphate receptors (IP3Rs), which release Ca^2+^ from the ER lumen into the cytosol [25,26,27].

In response to physiological (or pathological) stress, such as proteotoxic stress, oxidative stress, or disturbances in Ca^2+^ homeostasis, an accumulation of unfolded proteins in the ER triggers the ER stress response. The PQC system in the ER is able to detect the accumulation of misfolded/unfolded proteins and, subsequently, activates the UPR^ER^, a major ER stress-responsive pathway that inhibits protein synthesis and expands ER folding capacity and degradation capacity. The UPR^ER^ consists of three stress-responsive arms: activating transcription factor 6 (ATF6), inositol-requiring transmembrane kinase/endoribonuclease 1 (IRE1), and protein kinase RNA-like endoplasmic reticulum kinase (PERK) (Figure 1). Under normal circumstances, these three transmembrane proteins are rendered inactive through the binding of GRP78. However, during cellular stress, GRP78 dissociates from ATF6, IRE1, and PERK, thereby activating them. Activated IRE1 splices *X-box binding protein 1* (*XBP1*) mRNA and the spliced XBP1 translocates to the nucleus where it acts as a transcription factor to initiate transcription of molecular chaperones and folding catalysts. ATF6 is cleaved in the Golgi upon ER stress activation and the N-terminal fragment acts as a transcription factor at protective response genes, including those that protect against ER stress and reactive oxygen species (ROS). Activated PERK phosphorylates eukaryotic initiation factor 2 alpha (eIF2α), resulting in a complete blockage of protein translation and initiation, except for the molecular chaperones and catalysts initiated by IRE1 and ATF6, which mRNAs consist of open reading frames in the 5′ untranslated region. In addition, eIF2α phosphorylation induces translation of ATF4, which is able to induce the expression of C/EBP homologous protein (CHOP) (Figure 1). The UPR^ER^ is initially an adaptive pathway to restore ER homeostasis, but, if unresolved, chronic/severe activation of the UPR^ER^ leads to cell dysfunction and apoptotic cell death, which is initiated by CHOP [6,28,29]. Prolonged ER stress may not be alleviated and cardiomyocytes may undergo apoptosis, which is in any other organ beneficial to clear irreparable damage. However, as cardiomyocytes do not regenerate, apoptosis is detrimental in the heart and prolonged ER stress must be avoided. On the other hand, mild cardiac ER stress may be beneficial to clear unfolded, damaged or old proteins, thereby maintaining proper heart function.

Thus, the ER stress-related signaling pathways activate transcriptional and translational mechanisms that reduce global protein synthesis, increase ER protein-folding capacity, and promote the degradation of misfolded proteins, thereby maintaining normal cardiac health and function.

## 3. Role of Mitochondria in Cardiac Health

Mitochondria are important organelles, especially in the heart, as they are the source of energy (adenosine triphosphate, ATP) and are required for numerous essential metabolic processes to maintain cardiac contractility and normal heart function [30]. The protein-folding environment in mitochondria is challenged by the complex organelle architecture and the delicate process of assembly of the electron transport chain. In addition, due to their role in energy production, mitochondria undergo continuous additional challenges, such as the management of ROS and the balance in potential mitochondrial DNA mutations [31].

The PQC system of mitochondria ensures mitochondrial integrity and proper mitochondrial function, thereby meeting the metabolic and functional demands of the cell. In response to physiological (or pathological) stress, the accumulation of misfolded/unfolded proteins in the mitochondria activates the UPR^mt^, initiating the transcription of nuclear-encoded mitochondrial proteases (ClpP), mitochondrial chaperones (HSP60, HSP10) and proteins involved in ROS detoxification and mitochondrial import, thereby ensuring the functional integrity of the mitochondrial proteome [15,32,33]. The exact mechanism of the UPR^mt^ is still somewhat elusive, but two pathways have been described (Figure 1). The first pathway comprises of the transcription factor ATF5, which contains both a nuclear and a mitochondrial targeting sequence. Under physiological conditions, ATF5 is imported into the mitochondria, where it is degraded by the protease LON. However, mitochondrial stress hampers the import of ATF5, which is consequently targeted to the nucleus, where it initiates UPR^mt^-associated transcription [30,34,35]. The second pathway consists of c-Jun N-terminal kinase 2 (JNK2) and PERK. In accordance with the UPR^ER^, PERK phosphorylates eIF2α, consequently blocking protein translation and initiation. Phosphorylation of eIF2α initiates the translation of ATF4, CHOP, and ATF5. In addition, JNK2 binds to the transcription factor c-Jun, which activates the transcription of CHOP. ATF4, CHOP, and ATF5 all initiate UPR^mt^-associated transcription (Figure 1) [30,35].

Mitophagy, a specialized form of autophagy, is also activated to safeguard mitochondrial proteostasis in response to mitochondrial stress [36]. It serves to eliminate the most severely defective/damaged mitochondria, while the UPR^mt^ promotes stabilization and recovery of the repairable mitochondria.

Thus, the mitochondrial PQC system monitors protein integrity and prevents accumulation of damaged proteins in the mitochondria to maintain proper protein folding and clearance of misfolded proteins in cells, thereby conserving cardiac contractility and normal heart function.

## 4. Interactions between the ER and Mitochondria

Over the past years, it has been observed that the ER and mitochondria intensively interact with each other and interaction is a prerequisite for healthy cardiac function [37], As a consequence, loss of interaction fuels ageing and cardiac disease. Mitochondria are spatially and functionally organized in close contact with the ER and this contributes to mitochondrial uptake of Ca^2+^ released from the ER by IP3Rs, thereby providing the mitochondria with Ca^2+^ that is essential for ATP production [37,38]. During the early phases of ER stress, ER Ca^2+^ leak and increased ER–mitochondrial coupling lead to an elevated mitochondrial Ca^2+^ uptake to promote mitochondrial respiration and bioenergetics [39]. Ca^2+^ signaling between the two organelles is accomplished through mitofusin 2 (Mfn2), an essential physical tether between the ER and mitochondria (Figure 2) [40]. The importance of Mfn2 is exemplified by Chen et al., who showed that cardiac-specific ablation of Mfn2 decreased ER–mitochondrial tethering by 30%, leading to a decreased mitochondrial Ca^2+^ uptake, which hampers the response to physiological stress [40]. Besides Mfn2, several other proteins, including VAPB, PTPIP51, GRP75, VDAC1, BAP31, FIS1, and Pdzd8, are important for the tethering and interactions, such as Ca^2+^ exchange, lipid trafficking, apoptosis, autophagy, and mitochondrial fission and fusion, between the ER and mitochondria [41]. Another example to stress the importance of the structural and physiological interactions between the ER and mitochondria consists of stromal interaction molecule 1 (STIM1). STIM1 is an ER Ca^2+^ sensor, which is essential for normal cardiac homeostasis through the maintenance of ER Ca^2+^ levels (Figure 2). This has been demonstrated in cardiomyocyte-specific STIM1 knockout mice, which do not only show pronounced ER dilatation, but also smaller mitochondria and a disrupted mitochondrial network. Importantly, these mice have cardiac fibrosis and develop dilated cardiomyopathy [42], suggesting that the structural and physiological interactions between the ER and mitochondria in the heart are of utmost importance for normal and healthy cardiac function.

An important interaction between the ER and mitochondria involves the regulation of autophagy, a fundamental cellular pathway that is activated as a pro-survival pathway under physiological ER stress [43,44]. The IRE1 arm of the UPR^ER^ activates JNK and phosphorylates Bcl-2, leading to the dissociation of Bcl-2 from the autophagy-related protein Beclin-1 (Atg6), which is localized at both the ER and mitochondria. This results in the promotion of autophagy as a pro-survival mechanism in the early stages of ER stress (Figure 2) [45,46]. A specialized form of autophagy—mitophagy—is initiated at the mitochondria-associated ER membranes (MAMs), where PINK1 and Beclin-1 re-localize to promote ER-mitochondria tethering and autophagosome formation [47]. The other way around, it has been demonstrated that dysfunctional and/or damaged mitochondria are able to induce ER stress [48]. In addition, the importance of the ER–mitochondrial interactions in the autophagic process comes from the notion that disruption of the ER–mitochondrial contacts impairs the formation of the autophagosome [49]. These findings indicate that interactions between the ER and mitochondria are important for the activation of autophagy to eliminate damaged proteins and organelles, thereby maintaining healthy heart function.

Next to their role in autophagy activation, interactions between the ER and mitochondria can regulate apoptosis under physiological conditions [50]. Bak, known for its localization on the outer mitochondrial membrane where it induces apoptosis upon activation, can also be targeted to the ER, where it depletes Ca^2+^, induces caspase 12 cleavage, and, subsequently, induces apoptosis (Figure 2) [51]. During the early phase of ER stress, the ER and mitochondria induce an increase in mitochondrial metabolism that depends crucially upon ER–mitochondrial coupling and Ca^2+^ transfer, which, by enhancing cellular bioenergetics, establishes the metabolic basis for the healthy adaptation to ER stress [39]. In addition, the pro-survival protein HCLS1-associated protein X-1 (Hax1), a novel endogenous inhibitor of apoptosis that is located at both the ER and mitochondria, is associated with maintaining a healthy mitochondrial function and protection from apoptotic cell death (Figure 2). Hax1 is significantly downregulated in cardiac cells upon ER stress, which also resulted in a disrupted mitochondrial morphology, Mfn2 downregulation, and ROS production. Importantly, overexpression of Hax1 protected against ER stress-induced apoptosis and mitochondrial changes, suggesting that Hax1 could be a critical modulator in the cross talk between the ER and mitochondria [52]. The tumor suppressor protein p53 is also an inducer of apoptosis (Figure 2). It can either act indirectly, as a transcription factor increasing or decreasing pro-apoptotic and anti-apoptotic proteins, respectively, or directly, by permeation of the outer mitochondrial membrane through interaction with Bcl-2, thereby releasing cytochrome c from the mitochondria [53,54]. However, p53 is also able to localize to the ER, where it contributes to the imbalance of ER–mitochondrial interactions by inducing mitochondrial damage, including the reduction of oxidative phosphorylation and the release of cytochrome c, leading to apoptotic cell death [55].

To summarize, the functional and balanced interactions, including Ca^2+^ handling, contractile function, autophagy, and apoptosis, between the ER and mitochondria contribute to the maintenance of cardiomyocyte homeostasis and cardiac contractile function.

## 5. Interactions between the ER and Mitochondria in Cardiac Ageing and Cardiac Disease

There is compelling evidence that alterations in the interactions between the ER and mitochondria play fundamental roles in the development and progression of cardiac ageing and disease, including ischemic heart disease, AF, and cardiomyopathy.

### 5.1. Alterations of Interactions between the ER and Mitochondria in Cardiac Ageing

Ageing is a primary risk factor for cardiac disease [56]. Cardiac ageing manifests as a decline in structure and function of the heart, leading to cardiac disease [57,58]. At the cellular level, ageing entails a decline in mitochondrial function and dysregulation of cellular processes, such as oxidative stress, autophagy, and metabolic imbalance, in cardiomyocytes [57]. Mitochondrial alterations, including impaired metabolism and metabolic flexibility with a decreased capacity to oxidize fatty acids and enhanced dependence on glucose metabolism, are critically involved in increased sensitivity to stress in the aged heart [59]. Interestingly, mitochondrial Ca^2+^ uptake is reduced in cardiomyocytes from aged hearts, and this effect is closely linked with decreased NAD(P)H production and increased mitochondrial ROS production upon increased contractile activity [60]. Moreover, a defective communication between the mitochondrial voltage-dependent anion channel (VDAC) and the SR RyR in cardiomyocytes from aged hearts is associated with a dysregulated Ca^2+^ handling. Age-dependent alterations in SR Ca^2+^ transfer to mitochondria and in Ca^2+^ handling in the SR and mitochondria can be reproduced in cardiomyocytes from young hearts after disrupting the connection between the SR and mitochondria, suggesting that defects in physiological interactions between the SR and mitochondria underlie inefficient inter-organelle Ca^2+^ exchange, and as such contributes to impaired mitochondrial function and energy demand–supply mismatch in the aged heart. In addition, it has been identified that the protective adaptive response of the UPR^ER^ is significantly reduced and pro-apoptotic signaling is more robust during ageing in several tissues, including brain, lung, liver, kidney and heart [61,62]. Moreover, the key quality control proteins, chaperones, and enzymes within the ER, such as GRP78, PDI, calnexin, and glucose-regulated protein 94 (GRP94), are impaired during the aging process [63]. Especially chaperones are progressively oxidized with age, contributing to their functional decline, which is in line with the observed impairment of an adequate cellular response to ER stress during cardiac ageing [63]. Impaired contractility and increased ER stress-induced apoptosis were observed in the heart of mice with cardiac-specific knockout of Sirtuin 1 (Sirt1), an NAD^+^-dependent histone deacetylase, suggesting that a Sirt1 activator could be a potential modulator of the ER stress response to protect the ageing heart through the inhibition of ER stress-induced apoptosis [64].

These findings indirectly indicate that functional interactions between the ER and mitochondria in the aged heart are hampered and contribute to impairment of autophagy, apoptosis, and mitochondrial dysfunction, leading to cardiac dysfunction during ageing. Although accumulating evidence for loss of the interactions between the ER and mitochondria underlies cardiac ageing, our knowledge on the molecular mechanisms involved is still limited and further research is warranted to conclusively elucidate the importance of loss of interactions during ageing.

### 5.2. Alterations of Interactions between the ER and Mitochondria in Myocardial Ischemia/Reperfusion

In the clinic, acute myocardial ischemia/reperfusion (I/R) underlies coronary heart disease (CHD). CHD typically arises in patients presenting with an acute myocardial infarction (MI), a major cause of death and disability worldwide [65]. Over the last decade, accumulating evidence shows that alterations in the interactions between the ER and mitochondria contribute to the onset and progression of cardiac I/R injury, in particular interactions involving Ca^2+^ handling, autophagy, and apoptosis [66,67]. In regards to Ca^2+^ handling, upon I/R stress, IP3R expression on the ER is upregulated, leading to Ca^2+^ overload in mitochondria, which, subsequently, activates apoptotic signaling in re-perfused hearts [68,69]. However, returning the ER–mitochondrial interaction back to basal conditions, via the reduction of the upregulated IP3R1 Ca^2+^ channel complex, decreases the Ca^2+^ transfer from the ER to mitochondria, subsequently attenuating mitochondrial Ca^2+^ overload in adult mouse cardiomyocytes during hypoxia-reoxygenation (HR). This suggests that balancing the structural and physiological interactions between the ER and mitochondria during reperfusion could protect cardiomyocytes from lethal reperfusion injury [67]. Thioredoxin-related transmembrane protein 1 (TMX1), a novel SERCA-inhibiting protein, is enriched on the MAMs, which is the site of the ER-mitochondria Ca^2+^ flux. Cancer cells with low TMX1 levels show increased ER Ca^2+^ levels, accelerated cytosolic Ca^2+^ clearance, and reduced Ca^2+^ transfer to mitochondria, indicating that inhibition of TMX1 may reduce the susceptibility of the heart to I/R injury via reducing MAM contacts [70]. In regards to autophagy, Atg14, a pre-autophagosome/autophagosome protein marker, is re-localized to the MAMs after starvation, and is an indispensable factor for mitophagy activation [71]. However, Atg14 is decreased in a cardiac HR model, resulting in impaired mitophagy and increased cardiomyocyte apoptosis [72]. Finally, in regards to apoptosis, the inhibition of ER stress in in vitro and in vivo myocardial I/R models reveals beneficial effects on cardiac injury, myocardial apoptosis, and infarct area. Inhibition of ER stress increases the expression of superoxide dismutase (SOD) and Bcl-2, which, consequently, decreases the expression of Bax, resulting in the inhibition of apoptosis [73].

These findings suggest that the functional imbalance in the interactions between the ER and mitochondria contribute to increased myocardial apoptosis and damage in cardiac I/R injury.

### 5.3. Alterations of Interactions between the ER and Mitochondria in AF

AF is the most common progressive cardiac tachyarrhythmia, and is associated with high morbidity and mortality worldwide [3]. There are strong indications that defective interactions between the ER and mitochondria may contribute to AF pathogenesis. In experimental AF models and AF patients, ER stress is present, which induces autophagy and, thereby, cardiac remodeling [74]. Importantly, overexpression of the ER chaperone GRP78 or prevention of ER stress by treatment with the chemical chaperone 4-PBA protected against cardiac remodeling and AF in experimental cardiomyocytes, *Drosophila melanogaster,* and a dog model for AF [74]. Moreover, ER stress may activate mitogen-activated protein kinases (MAPKs), which initiate the mitochondrial apoptotic pathway [75]. In addition, atria of patients with persistent AF show oxidized RyR2, the cardiac SR Ca^2+^ release channel. Oxidized RyR2 leads to aberrant intracellular Ca^2+^ release (Ca^2+^ sparks) and promotes the development of AF, suggesting that defective communication between the SR and mitochondria contributes to the progression of AF [76]. In concordance, experimental AF models and AF patients show mitochondrial stress, exemplified by increased expression of HSP60 and HSP10, decreased ATP expression, loss of the mitochondrial membrane potential, and mitochondrial network fragmentation, resulting in contractile dysfunction and AF progression [77]. These observations suggest that ER stress may be linked with mitochondrial stress, and that an imbalance in the functional interactions between the ER and mitochondria may contribute to AF progression.

In summary, there is proof for a key role of ER stress and mitochondrial stress in AF. Whether this is a result of loss of interactions between the ER and mitochondria is not known.

### 5.4. Alteractions of Interactions between the ER and Mitochondria in (Inherited) Cardiomyopathy

Accumulating evidence indicates that ER stress and mitochondrial stress play important roles in cardiomyopathy [78,79,80], of which approximately 50% is inherited [81]. Elevated expression of the ER stress markers GRP78, eIF2α, and XBP1 and increased activation of the UPR^ER^ is observed in patients with the p.S143P mutation in the intermediate filament gene *lamin A/C*, the most frequently reported genetic variant in inherited dilated cardiomyopathy (DCM) [82]. Enhanced mitochondrial oxidative stress and mitochondrial dysfunction were observed in the heart of cats with hypertrophic cardiomyopathy (HCM) [83]. Moreover, impaired autophagy was shown in a *Mybpc3*-targeted knockin HCM mouse model and in DCM patients with mutations in *PLEKHM2* [84,85]. Although ER stress and mitochondrial stress are undeniably present in different forms of cardiomyopathy, it is, however, not known whether loss of the functional interactions between the ER and mitochondria affect the pathogenesis of cardiomyopathies. Therefore, further research is needed to unravel how the molecular mechanisms in the regulation of the interactions between the ER and mitochondria contribute to cardiomyopathies.

### 5.5. Alterations of Interactions between the ER and Mitochondria in Diabetic Cardiomyopathy

Diabetic cardiomyopathy is characterized as dysfunctional remodeling of the myocardial structure and abnormal cardiac performance in patients with diabetes mellitus, which is independent of vascular pathology [86]. Increasing evidence shows that alterations in the interactions between the ER and mitochondria are involved in the pathophysiological process of diabetic cardiomyopathy. Reduced protein levels of RyR2 in streptozotocin (STZ)-induced diabetic rats contributed to a decrease in the SR Ca^2+^ storage and decreased rates of Ca^2+^ release in cardiomyocytes [87], suggesting that reduced contacts between the SR and mitochondria and the subsequent aberrant intracellular Ca^2+^ signaling may contribute to diabetic cardiomyopathy. On the other hand, reduced contacts between the SR and mitochondria may be beneficial, as the downregulation of Mfn2 expression in STZ-induced type I diabetes inhibited the interaction between the ER and mitochondrial apoptotic pathways [88]. More interestingly, the expression of FUN14 domain containing 1 (Funcd1), an outer mitochondrial membrane protein important for mitophagy and MAMs, was increased in cardiac tissue from diabetic patients. In STZ-induced diabetic mice, increased Funcd1 induced MAM formation, increased mitochondrial Ca^2+^ influx and, subsequently, mitochondrial dysfunction, resulting in diabetic cardiomyopathy [89]. These observations suggest that the structural and functional interactions between the ER and mitochondria have a fine line between being beneficial and being detrimental and should, therefore, be precisely regulated. In experimental diabetic cardiomyopathy, severe ER stress and prolonged UPR^ER^ activation contribute to cardiomyocyte apoptotic death [90,91]. In line, markers of ER stress (GRP78 and CHOP) and apoptosis (cleaved caspase-3) were elevated in myocardium of diabetic patients [92].

Taken together, an imbalance in the interactions between the ER and mitochondria may contribute to mitochondrial dysfunction and lead to cardiomyocyte apoptotic death in diabetic cardiomyopathy.

## 6. ER and Mitochondria Interactions as Therapeutic Targets

As discussed above, alterations in and loss of the interactions between the ER and mitochondria may induce cardiomyocyte dysfunction and cardiac disease. Therefore, therapeutic interventions to improve interactions between the ER and mitochondria may conserve cardiac function and represent possible promising strategies to delay cardiac ageing and to delay or prevent cardiac disease.

### 6.1. Lifestyle Interventions—Caloric Restriction and Exercise

Lifestyle change by caloric restriction (CR) is one of the most effective nutritional interventions that reproducibly showed protection against cardiac risk factors and age-related cardiac disease in humans [93]. Besides CR, also exercise training is an effective non-pharmacological approach to improve heart function in cardiac ageing and disease [6]. In an aged rat model, a combination of intermittent ladder-climbing exercise training and a reduced caloric intake were found to decrease the levels of ER stress-related proteins, including phosphorylated PERK and CHOP, proteins that contribute to cardiac muscle damage in ageing [94]. Moreover, high-intensity training can improve cardiac function and reduce cardiac infarction by decreasing the expression of GRP78, phosphorylated PERK, phosphorylated eIF2α, ATF4, ATF6, XBP1, CHOP, and cleaved caspase-3 in an intermittent I/R rat model [95,96]. In addition, treadmill exercise has been shown to ameliorate ER stress by down-regulating phosphorylated eIF2α and ATF6 in diabetic mice [97]. As ageing is an important risk factor for cardiac disease, including AF, lifestyle interventions such as CR and proper exercise training could prevent ER stress, mitochondrial stress, and the progression of AF [98]. However, the effect of CR and exercise training on ER–mitochondrial interactions is so far unknown.

### 6.2. Pharmacological Interventions—Improved Interactions between the ER and Mitochondria

Pharmacological therapies that improve the balance in interactions between the ER and mitochondria attenuate ER and mitochondrial stress, thereby restoring heart function in ageing and cardiac disease.

One option to improve the ER–mitochondrial balance is by means of taurine, a marketed key nutrient for cardiac health. In ischemic rat cardiomyocytes, administration of taurine attenuated ER stress (GRP78, ATF6, PERK), mitochondrial oxidative stress, and mitochondrial-dependent apoptosis (Bax, Bcl-2) [99]. Another option is the utilization of metformin, a traditional anti-diabetic drug, which decreases ER stress-induced cardiac injury through the protection of mitochondria and attenuation of CHOP expression during cardiac I/R [100]. Moreover, sulodexide (SDX), a glycosaminoglycan, decreases cardiac injury, infarct area, and myocardial apoptosis during I/R through increased Bcl-2 and decreased Bax expression in a mouse model of I/R, suggesting that SDX has in vivo a cardioprotective role in the suppression of ER stress and apoptosis [73]. In addition, ER–mitochondrial cross talk can be regulated by exogenous H2S, which inhibits the activation of apoptotic pathways [88]. Finally, the administration of edaravone, an antioxidant, proved to be beneficial in ameliorating oxidative and ER stress-mediated myocardial apoptosis in an experimental DCM model [101].

Taken together, several compounds may improve the interactions between the ER and mitochondria, thereby having a beneficial effect on cardiac ageing and disease. However, further research is needed to elucidate compounds that directly target the ER–mitochondrial interactions. Nevertheless, with the current knowledge, a combination of lifestyle interventions (caloric restriction and/or high intensity training) and a pharmacological intervention, taurine, may be the most beneficial to improve the ER–mitochondrial interactions in cardiac disease.

## 7. Conclusions

A healthy proteostasis in cardiomyocytes maintains the proper contractile function in the heart. The ER and mitochondria are key players in the regulation of protein homeostasis and are important for the clearance of stress-induced misfolded proteins, thereby guaranteeing cardiomyocyte health. The structural and functional interactions between the ER and mitochondria are essential for normal cardiac function. Loss of these interactions contribute to the progression of ageing and cardiac disease and preservation of the ER–mitochondrial interactions by pharmacological targeting may represent a promising strategy to conserve cardiac function.

## Figures and Tables

**Figure 1 cells-08-01617-f001:**
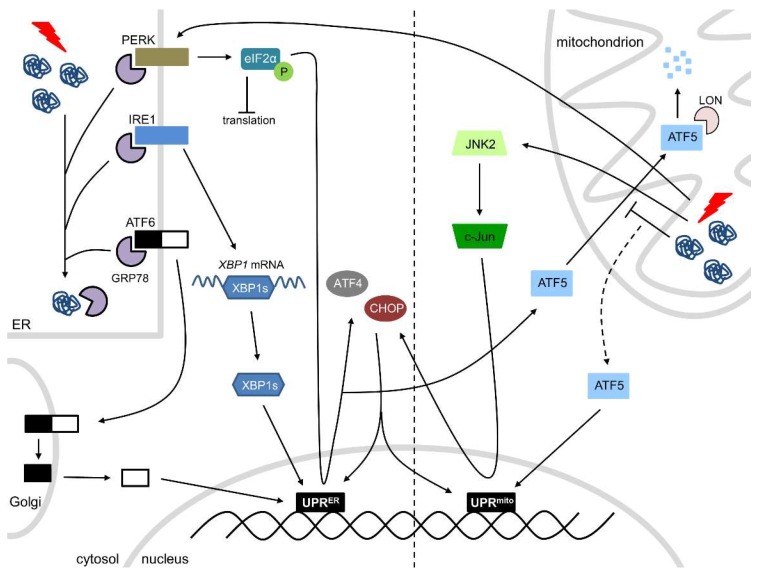
The ER and mitochondrial UPRs. ER stress activates the three arms of the UPR^ER^: PERK, IRE1, and ATF6. PERK activation leads to phosphorylation of eIF2α, resulting in a protein translation block and transcription of ATF4 and CHOP. IRE1 splices *XBP1* mRNA and a spliced form translocates to the nucleus. ATF6 is spliced in the Golgi and the N-terminal fragment acts as a transcription factor. All three arms initiate the transcription of ER-related molecular chaperones and/or folding catalysts. Mitochondrial stress activates the UPR^mt^, which consists of ATF5, PERK, and JNK2. During mitochondrial stress, the import of ATF5 into the mitochondria is blocked, leading to the translocation of ATF5 to the nucleus. PERK activation leads to the transcription of ATF4, CHOP, and ATF5. JNK2 binds to the transcription factor c-Jun, which activates the transcription of CHOP. ATF5, PERK and JNK2 all initiate the transcription of mitochondrial proteases, mitochondrial molecular chaperones, and proteins involved in ROS detoxification and mitochondrial import.

**Figure 2 cells-08-01617-f002:**
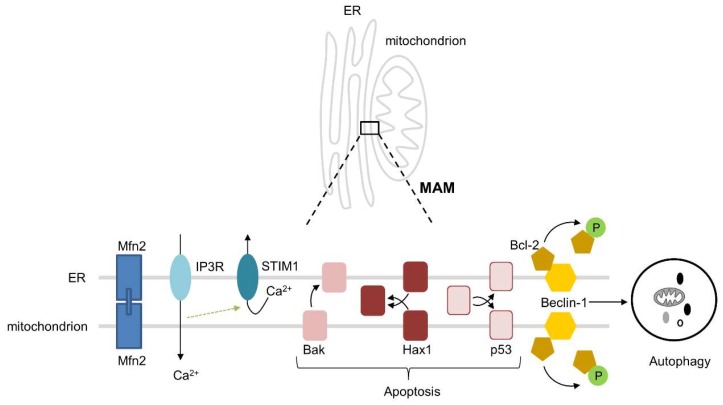
The interactions between the ER and mitochondria. Mfn2 is located on both the ER and mitochondrial membranes and is a physical tether between these two organelles. IP3Rs on the ER release Ca^2+^, which is taken up by the mitochondria, thereby providing the Ca^2+^ that is necessary for ATP production by the mitochondria. STIM1 is an ER Ca^2+^ sensor, which restores the Ca^2+^ storage in the ER upon Ca^2+^ release. Bak, Hax1, and p53 can all initiate apoptosis. Bax is located on the mitochondrial membrane and upon translocation to the ER membrane initiates apoptosis. Hax1 is located on both the ER and mitochondrial membranes, where it protects against apoptosis initiation. When p53 translocates to either the ER or mitochondrial membrane, apoptosis is activated. Beclin-1 is localized at both the ER and mitochondria. Upon phosphorylation of Bcl-2, this protein dissociation from Beclin-1, thereby activating autophagy.

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
