# Peer review of "Imbalance of ER and Mitochondria Interactions: Prelude to Cardiac Ageing and Disease?"

_cells, 2019, doi:10.3390/cells8121617_

Round 1
Reviewer 1 Report
This manuscript examines the imbalance of ER and mitochondria interactions that are involved in cardiac ageing and disease. This is a very important and timely topic and presents an area where improvements in understanding may provide future options for therapeutic targets.
Comments:
Please spell out atrial fibrillation in the abstract. In line 68, for completeness, it would be good to indicate what the one-third of newly synthesized proteins are which are made processed by the ER. In line 93, perhaps the wording “ATF6 is cleaved in the Golgi upon activation…” could be changed to “ATF6 is cleaved in the Golgi upon ER stress...”. In lines 93-94, the authors state that “ATF6 is cleaved in the Golgi upon activation and the N-terminal fragment acts as a transcription factor for additional ER-related molecular chaperones.” Nevertheless, the adaptive roles of ATF6 in the heart extend beyond the induction of ER chaperones. The authors should revise this topic accordingly. In line 97, the authors state that “eIF2α induces transcription of ATF4”. Phosphorylation of eIF2α results in ATF4 translation. Did the authors mean translation? Same for lines 142-143. In lines 101-103, the authors end the paragraph with “Under physiological conditions, prolonged ER stress may not be alleviated and cardiomyocytes may undergo apoptosis, which is beneficial to clear irreparable damage, thereby maintaining proper heart function under mild stress conditions.” What do the authors mean by “under physiological conditions, prolonged ER stress”? Also, “prolonged ER stress” and “mild ER stress” do not go together. Perhaps this sentence should be revised. It is unclear what the authors actually mean. Moreover, does it make sense that in the heart, cardiomyocyte death would be beneficial? Regarding Figure 1, please consider decreasing the amount of white space in between. It would be good to use more of the white space to actually make the colorful shapes and the names of the proteins larger. If you make the arrows smaller, that would also decrease the white space. Are there any citations (reviews maybe) that can be used for the first sentence of the paragraph, on lines 153-154? It is not clear what the authors mean by “direct” and “indirect” categorization of the ER and mitochondria interactions. Furthermore, since this is more of a subjective categorization, and more likely all ER events impact on mitochondrial events and vice versa, the authors should consider revising this section with regards to direct and indirect. In figure 2, the diagram of the ER and mitochondria is too light to be seen. Please use a darker color. Also, please reduce the large white spaces. In line 231, the comma before cardiomyocytes should be removed. The authors should clarify if lines 244-245 refers to the heart or another system. In line 270, “adult mice cardiomyocytes” should be changed to “adult mouse cardiomyocytes.” Please define MAM. In line 268- 277, the authors refer to a reduction of the ER and mitochondrial interaction as being beneficial. However, in Section 5.3, they state that the loss of interactions happens in AF. Can you make sure to clear up this inconsistency? Perhaps there is a way to make this clearer. It is the opinion of the reviewer that the studies reviewed under Section 5.3 do not actually illustrate loss of interaction between ER and mito. Some of these might be altered interactions between ER and mito. In line 316, “undeniable” should be changed to “undeniably”. ER and mitochondria interactions as therapeutic target should be changed to ER and mitochondria interactions as therapeutic targets. Based on its effects on which of the proteins (interactors) that the authors reviewed does taurine improve interactions of ER and mito? The authors should discuss which of the interactors that they describe in the review are the most likely and important ones to target for therapies and in which cardiac disease settings. This expert view would help to move the field forward and focus future studies.
Author Response
We would like to thank reviewer 1 for his/her comments and feedback. We have adapted the manuscript accordingly as suggested by this reviewer and outlined below the major changes.
Lines 67-70: ‘The ER lumen constitutes of a specialized folding environment, including ER chaperones and oxidoreductases, which processes approximately one-third of newly synthesized proteins, including secretory, cell membrane and ER proteins, to ensure that they are correctly folded and assembled’ Lines 93-95: ‘ATF6 is cleaved in the Golgi upon ER stress activation and the N-terminal fragment acts as a transcription factor at protective response genes, including those that protect against ER stress and ROS.’ Line 105-108: ‘However, as cardiomyocytes do not regenerate, apoptosis is detrimental in the heart and prolonged ER stress must be avoided. On the other hand, mild cardiac ER stress may be beneficial to clear unfolded, damaged or old proteins, thereby maintaining proper heart function.’ Line 255-257: ‘In addition, it has been identified that the protective adaptive response of the UPRER is significantly reduced and pro-apoptotic signaling is more robust during ageing in several tissues, including brain, lung, liver, kidney and heart.’ To clarify the inconsistency between reductions in the ER and mitochondrial interactions being beneficial or detrimental, we changed the sentence in lines 280-283: ‘However, returning the ER-mitochondrial interaction back to basal conditions, via the reduction of the upregulated IP3R1 Ca2+ channel complex, decreases the Ca2+ transfer from the ER to mitochondria, subsequently attenuating mitochondrial Ca2+ overload in adult mouse cardiomyocytes during hypoxia-reoxygenation (HR).’ We adapted the sentence about taurine and incorporated the proteins that taurine improves (line 383-385): ‘In ischemic rat cardiomyocytes, administration of taurine attenuated ER stress (GRP78, ATF6, PERK), mitochondrial oxidative stress and mitochondrial-dependent apoptosis (Bax, Bcl-2).’ As suggested by the reviewer, we included a sentence with which interactors are the most important to target for therapies (line 398-400): ‘Nevertheless, with the current knowledge, a combination of lifestyle interventions (caloric restriction and/or high intensity training) and a pharmacological intervention, taurine, may be the most beneficial to improve the ER-mitochondrial interactions in cardiac disease.’Reviewer 2 Report
The review manuscript by Li et. al. on the imbalance of ER and mitochondrial interactions during the cardiac aging and diseases is very well written comprehensive work. Manuscript can be accepted for publication.
Comments:
In this review article, the authors focused on the ER and mitochondrial interactions during cardiac aging and diseases. As authors highlighted, cardiac diseases are still a major health concern worldwide. Recent studies suggest that the imbalance in protein quality control plays a crucial role in the onset of multiple cardiac pathologies. On the other hand, mitochondria, which occupy around 30% of cardiomyocytes volume, match up the high energy demand of the heart. Any mitochondrial dysfunction leads to the metabolic catastrophe in the heart and, ultimately heart failure. The ER-mitochondrial interactions are transient but crucial for the protein, lipid, ions exchange between these two crucial cellular organelles. In the current review article, authors have covered various aspects of the ER-mitochondrial contacts, specifically the protein quality control aspect, in terms of cardiac health, which will be essential to understand how ER-mitochondria crosstalk may be beneficial in designing novel therapeutic agents. The manuscript is well written and presented in a logical manner. The reviewer has some minor suggestions which can be incorporated to improve the manuscript quality.
Specific comments:
1.In section 4: “interactions between the ER and mitochondria” authors cited a review article for the role of ER-mitochondrial contacts and Ca2+ transport, which is essential for the ATP production (Ref# 37). It would be great if authors can cite some original work.
2.In Section 4: authors mention the involvement of the Mfn2 in ER-mitochondrial contacts, it would be great if authors can detail the molecular determinants for the ER-mitochondrial contacts. Please discuss the role of Grp75, Bap31, Pdzd8 and other known modulators of the ER-mitochondrial contacts along with their possible role in cardiac health and disease.
3.In Section 4: authors discuss the role of ER-mitochondrial contacts in autophagy induction and have not discussed the major findings from the Lippincott-Schwartz Lab (Cell. 2010 May 14;141(4):656-67. doi: 10.1016/j.cell.2010.04.009) where they described the role of ER-mitochondrial contacts in the lipids transport during the autophagosome formation. Please discuss the role of ER-mitochondrial contacts in terms of lipid trafficking and its involvement in cardiac health and disease.
Author Response
We would like to thank reviewer 2 for his/her comments and feedback. We have adapted the manuscript accordingly as suggested by this reviewer.
We have added a citation of original work in section 4, in addition to the review article about the role of ER-mitochondrial contacts and Ca2+ transport: Rizzuto R, Brini M, Murgia M, Pozzan T. Microdomains with high Ca2+ close to IP3-sensitive channels that are sensed by neighboring mitochondria. Science. 1993;262:744. The following sentence was added (line 169-172) to indicate that Mfn2 is not the only protein involved in the ER-mitochondrial interactions: Besides Mfn2, several other proteins, including VAPB, PTPIP51, GRP75, VDAC1, BAP31, FIS1 and Pdzd8, are important for the tethering and interactions, such as Ca2+ exchange, lipid trafficking, apoptosis, autophagy and mitochondrial fission and fusion, between the ER and mitochondria.’ We incorporated a sentence explaining the role of the ER-mitochondrial in autophagosome formation (line 190-192): ‘In addition, the importance of the ER-mitochondrial interactions in the autophagic process comes from the notion that disruption of the ER-mitochondrial contacts impairs the formation of the autophagosome.’